# A new phylodynamic model of *Mycobacterium bovis* transmission in a multi-host system uncovers the role of the unobserved reservoir

**Anthony O'Hare**[1], **Daniel Balaz**[2], **David M. Wright**[3,4], **Carl McCormick**[3],
**Stanley McDowell**[3], **Hannah Trewby**[5], **Robin A. Skuce**[3], **Rowland R. Kao**[2]*

**1** Computing Science and Mathematics, Faculty of Natural Sciences, University of Stirling, Stirling, United Kingdom, **2** Royal (Dick) School of Veterinary Studies and The Roslin Institute, University of Edinburgh, Edinburgh, United Kingdom, **3** Veterinary Sciences Division, Agri-Food and Biosciences Institute, Stormont, Belfast, Northern Ireland, United Kingdom, **4** School of Biological Sciences, Queen's University Belfast, Belfast, Northern Ireland, United Kingdom, **5** Quadram Institute. Norwich, Norfolk, United Kingdom

* Rowland.Kao@ed.ac.uk

**Data Availability Statement:** The genomic data is already publicly available on genbank having been previously published in Trewby et al. 2016 (doi: 10.

## Abstract

Multi-host pathogens are particularly difficult to control, especially when at least one of the hosts acts as a hidden reservoir. Deep sequencing of densely sampled pathogens has the potential to transform this understanding, but requires analytical approaches that jointly consider epidemiological and genetic data to best address this problem. While there has been considerable success in analyses of single species systems, the hidden reservoir problem is relatively under-studied. A well-known exemplar of this problem is bovine Tuberculosis, a disease found in British and Irish cattle caused by *Mycobacterium bovis*, where the Eurasian badger has long been believed to act as a reservoir but remains of poorly quantified importance except in very specific locations. As a result, the effort that should be directed at controlling disease in badgers is unclear. Here, we analyse densely collected epidemiological and genetic data from a cattle population but do not explicitly consider any data from badgers. We use a simulation modelling approach to show that, in our system, a model that exploits available cattle demographic and herd-to-herd movement data, but only considers the ability of a hidden reservoir to generate pathogen diversity, can be used to choose between different epidemiological scenarios. In our analysis, a model where the reservoir does not generate any diversity but contributes to new infections at a local farm scale are significantly preferred over models which generate diversity and/or spread disease at broader spatial scales. While we cannot directly attribute the role of the reservoir to badgers based on this analysis alone, the result supports the hypothesis that under current cattle control regimes, infected cattle alone cannot sustain *M. bovis* circulation. Given the observed close phylogenetic relationship for the bacteria taken from cattle and badgers sampled near to each other, the most parsimonious hypothesis is that the reservoir is the infected badger population. More broadly, our approach demonstrates that carefully constructed bespoke models can exploit the combination of genetic and epidemiological data to overcome issues of extreme data bias, and uncover important general characteristics of transmission in multi-host pathogen systems.

1016/j.epidem.2015.08.003). The data on the demographics of cattle farms, including the movements of livestock between them, is subject to EU laws regarding personal disclosure and is subject to a data sharing agreement with the data holders, the Northern Ireland Dept. of Agriculture, Environment and Rural Affairs. Since the original agreement for these data was signed, the Northern Ireland Food Animal Information System (NIFAIS) has replaced Animal and Public Health Information System (APHIS). https://www.daera-ni.gov.uk/contacts/nifais-programme and is now the point of contact for further data requests. Requests for APHIS/NIFAIS data to be directed to: DAERA Veterinary Service Animal Health Group Information & Communication Branch, Ballykelly House, 111Ballykelly Road, Ballykelly, Limavady, BT49 9HP Email: vsinfo&commsbranch@daera-ni.gov.uk As used in this project, and according to our DSA it is recorded that: The dataset may be enriched with data on land use (using the CORINE land classification data set, which are available for use), and climate data (the Meteorological Office UK weather data, for which the AFBI veterinary epidemiology PI (A. Byrne) has a DSA. In this case, AFBI generated data from *M. bovis* culture and MLVA typing at animal-level from SICCT reactors and cattle found lesioned at routine slaughter (LRS) at animal-level under the DAERA AFBI-assigned Work Programme. APHIS records were 'enriched' by linking to these data and in order to replicate our results would require permission to link in the same fashion. The code implementing this model and requisite anonymised data required to reproduce our results in this paper can be found at https://github.com/anthonyohare/NIBtbClusterModel.

**Funding:** This study was funded by the following research grants: i) Biotechnology and Biological Sciences Research Council (BB/L010569/1) was awarded to RRK and funded the work of DB, DMW, TM, CM, SM, RAS (www.bbsrc.ac.uk) ii) Biotechnology and Biological Sciences Research Council (BB/L010569/2) was awarded to RRK and funded the work of DB iii) A Wellcome Trust Senior Fellowship (081696/Z/06/Z) was awarded to RRK and funded the work of AOH (www.wellcome.ac.uk) iv) Biotechnology and Biological Sciences Research Council (BB/P010598/1) was awarded to RRK and funded his involvement v) Dept of Agriculture environment and Rural Affairs grant DARD0407 was awared to RAS and funded the work of TM, CM, SM (https://www.daera-ni.gov.uk). The funders had no role in study design, data collection and analysis, decision to publish, or preparation of the manuscript.

## Author summary

For single host pathogens, pathogen genetic data have been transformative for understanding the transmission and control of many diseases, particulary rapidly evolving RNA viruses. However garnering similar insights where pathogens are multi-host is more challenging, particularly when the evolution of the pathogen is slower and pathogen sampling often heavily biased. This is the case for *Mycobacterium bovis*, the causative agent of bovine Tuberculosis (bTB) and for which the Eurasian badger plays an as yet poorly understood role in transmission and spread. Here we have developed a computational model that incorporates *M. bovis* genetic data from cattle only with a highly abstracted model of an unobserved reservoir. Our research shows that a model in which the reservoir does not contribute to pathogen diversity, but is a source of infection in spatially localised areas around each farm, better describes the patterns of outbreaks observed in a population-level sample of a single *M. bovis* genotype in Northern Ireland over a period of 15 years, compared to models in which either the reservoir has no role, disease spread is spatially extensive, or where they generate considerable diversity on their own. While this reservoir model is not explicitly a model of badgers, its characteristics are consistent with other data that would suggest a reservoir consisting of infected badgers that contribute substantially to cattle infection, but could not maintain disease on their own.

## Introduction

The analysis of high throughput genome sequence data for *Mycobacterium bovis* has already generated important insights into the relative roles of direct transmission and other mechanisms in the maintenance of bTB in cattle [1]. Central to this problem is the well-documented involvement of the Eurasian badger (*Meles meles*) in the persistence and spread of bTB. While it is known that badgers contribute to infection in cattle, the relatively poor and biased data available regarding their contribution mean that their importance to the problem remains poorly understood, a problem shared with many other multi-host pathogen systems. Previous analyses have built on small datasets, or used analytical tools based on evolutionary models (e.g. [2]) which, while providing useful insight [3–6], have only limited ability to exploit the much richer data available on the contact patterns recorded for the cattle population involved.

Here, we exploit these data in a agent-based simulation, using a partial-likelihood fitting approach based on a measure previously developed to fit animal-level transmission patterns to summary measures of herd outbreaks [7]. We compare models where the diversity patterns are generated (i) by cattle only, (ii) by cattle with a passive reservoir that produces minimal additional phylogenetic diversity and (iii) cattle plus an active reservoir, generating diversity consistent with the observed bacterial mutation rates. Considering also the spatial extent of reservoirs (locally around each farm or across all farms), we fit the models to a previously described *M. bovis* dataset [3]. Our analyses show a substantial preference for a model that includes a reservoir with only short range interactions, and consistent with (ii) above, transmits the most recently available genetic type back to the cattle population.

### Data

Northern Ireland has a well-developed test and slaughter program in which all cattle herds are tested for bTB on an annual basis. Since the 1990s, *M. bovis* isolates from infected cattle have been stored and typed using spoligotyping and more recently, combined with Variable

**Competing interests:** The authors have declared that no competing interests exist.

Number Tandem Repeat (VNTR) typing, to differentiate molecular types [8, 9]. Isolates are stored frozen and are available for re-culture to extract further DNA for sequencing. As described previously [3], a total of 145 VNTR-10 *M. bovis* isolates were included in this study, from 66 herd breakdowns (i.e. where at least one confirmed positive test occurred for a bovid in a herd previously considered bTB-free) in 52 herds between June 1995 and December 2010.

In Northern Ireland, detailed information on the cattle population, movements between herds and bTB test results is recorded by the Department of Agriculture, Environment and Rural Affairs [10] on the Animal and Public Health Information System (APHIS). All direct cattle movements between herds with VNTR-10 samples represented in our dataset (66 sequenced breakdowns and 12793 individual cattle movements) were made available, and were combined with the anonymised location and date of each sequenced sample. The locations (main farm building) of 52 herds that had a breakdown with VNTR-10 between 1995 and 2010 were made available from the APHIS database.

The data describing the location of the herds consist of the (anonymised via rotation, scaling and translation so that relative distances remained intact) x-y locations of 52 farms from which cultured samples were taken. The movement of cattle into or out of this network extended the number of farms in our dataset to 21,012.

Because herd population data were not available for the entire period over which bacterial samples were taken, we develop a parameterisation of the herd population relevant to our study and use this to populate our simulations. The size of each herd was recorded on the 1st of January and 1st of July of each year from 2003 to 2012 and we draw the initial herd size for our simulations from this distribution of herd sizes. The herd sizes in our available data were not found to change substantially over time so this simplifying assumption is not thought to substantially affect the results. Each herd is subjected to an annual whole herd test, and failing herds are subjected to both movement restrictions and follow-up whole herd tests until two successive clear tests are observed. From the record of herd tests on each farm, 403 herds had a whole herd test scheduled within 60 days of the start of the (recorded test) dataset. Assuming that each of these herds with tests scheduled within 2 months of the simulation start date were subject to follow-up tests, this represents ≈ 2% of farms under movement restriction. At the start of each simulation we set a similar number of herds under movement restrictions and to each herd we randomly assign the number of clear tests achieved at the simulation start time (either 0 or 1) to mimic the observed conditions. All other herds have routine tests scheduled every year (with the first test in the simulation scheduled at a random time within the first year of the start of the simulation), with each animal in the herd subjected to a test with sensitivity $\Omega$.

The phylogenetic dataset has previously been described in [3]. In brief, high-density whole genome sequencing (WGS) was performed on 145 (139 cattle and 6 badger) VNTR-10 isolates. Some badger isolates were available from a survey of badgers killed on the roads in Northern Ireland [11, 12]. However as these few isolates are likely to be only a small proportion of the total infected badgers in the area, are unlikely to be representative, and the locations of where the badger carcasses were found could not be verified, they were deemed unsuitable for inclusion in any further population-wide analysis (though we note the close phylogenetic relationship between the badger-derived samples and nearby ones from cattle [3]). Pairwise single nucleotide polymorphism (SNP) differences between sequenced samples were recorded and a histogram of these SNP differences generated as the basis of our further analysis, Fig 1.

## Methods

As in our previous study [7], we consider a simple four state model for the transmission of bTB where cattle are either susceptible ($S_C$), exposed ($E_C$), test-sensitive ($T_C$), or infectious

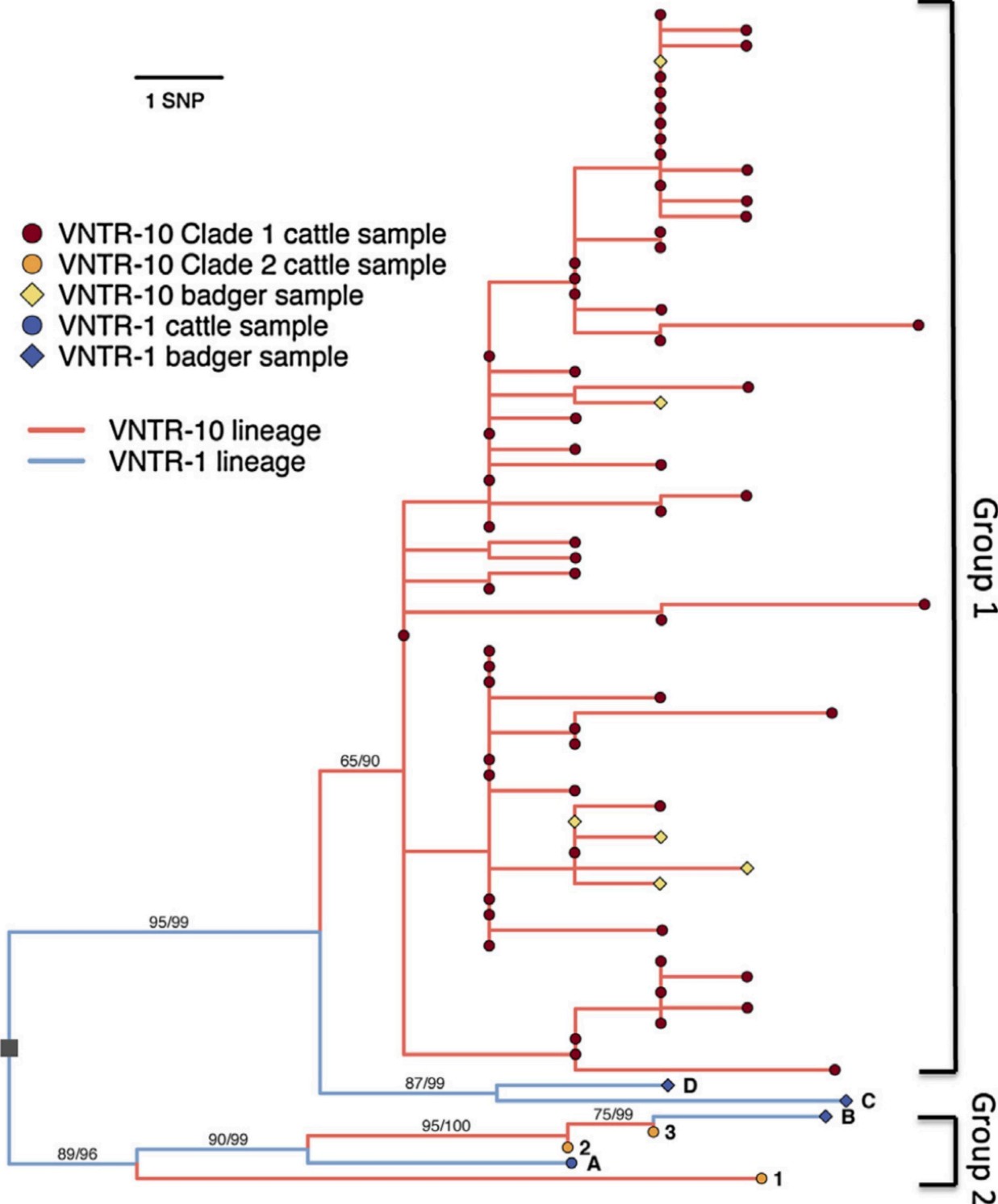

**Fig 1. Phylogeny of VNTR-1 and -10 isolates (from [3]) showing the distribution of SNPs in sampled cattle.** In this paper we have ignored the samples from badgers but show them here to indicate the close phylogenetic relatonship to the cattle isolates.

($I_C$). The tuberculin test, used in the UK, is based on a cow's response to the invading *M. bovis* and it is assumed that animals in the exposed stage have not yet mounted a sufficient response to be detected. Thus for the purposes of this model, any test on animals in this stage will return a negative result. Infections in both the test-sensitive and infectious stages are detectable. Once an animal becomes infectious, it remains so until it is detected, at which point the animal would be culled. In addition we create an infectious 'reservoir' population which generates an infection pressure and (potentially) generates additional genetic diversity in the bacterial population. Further we extend this to also consider susceptible ($S_r$) and infected ($I_r$) reservoir locations.

Once infected, animals progress from the exposed to test-sensitive stage at a rate $\sigma$ and from the test-sensitive state to infectious at a rate $\gamma$. Cattle-to-cattle transmission is at a rate $\beta S_C I_C$ (where $S_C$, $I_C$, are the numbers of cattle in these states) and we separate the cattle-reservoir and reservoir-cattle transmission rates as $\beta_{Cr}$ and $\beta_{rC}$ respectively.

We can write the model excluding cattle movements between farms as a set of Ordinary Differential Equations (ODEs) in the form

$$
\begin{aligned}
\dot{S}_C &= -\beta S_C I_C - \sum_r \beta_{rC} S_C I_r \\
\dot{E}_C &= \beta S_C I_C + \sum_r \beta_{rC} S_C I_r - \sigma E_C \\
\dot{T}_C &= \sigma E_C - \gamma T_C \\
\dot{I}_C &= \gamma T_C \\
\dot{I}_r &= \sum_F \beta_{Cr} S_r I_C
\end{aligned}
\tag{1}
$$

where the sums are over all reservoirs connected to each farm or all the farms connected to each reservoir. We assume that reservoir populations cannot infect each other. Also, once infected, reservoirs remain infected for the duration of the simulation. This could be interpreted, for example, as at least one individual within the reservoir remaining infected at any time once the reservoir has become infected. Animals in the $T_C$ and $I_C$ states are routinely tested and if detected, are removed. A schematic of this model can be found in the S1 Fig.

We solve the model using Gillespie's $\tau$-leap method [13] with a fixed $\tau$ of 14 days. In each 14 day step we calculate the numbers of animals whose disease status will have changed and update them accordingly, perform any whole herd tests (WHT), maintain a record of scheduled WHTs (including any follow-up tests from failed WHTs in the current period), move animals between farms, and update the transition kernel for the Gillespie algorithm in the next time step.

## Movements

To reduce computational costs, we only create agents for infected cows, as the number of cattle moved between farms is typically few compared to the herd size, and the susceptible fraction in a herd is high. We assume that animals born into the herd, replacing animals moved out of the herd and thus keeping the herd size constant, are susceptible and thus not tracked so we do not explicitly simulate birth and death processes. This simplifying assumption did not give results that differed significantly from less computationally efficient simulations where the size of each herd, births and deaths were explicitly tracked (not shown). As we do not know the force of infection associated with each reservoir we apply the assumption that it is constant

once the reservoir begins to harbour infection. We keep a record of each farm (herd) and its associated reservoirs in memory.

Every movement of cattle that passes through the VNTR-10 herds was recorded and used to create a distribution of farm-to-farm movements as well as a distribution of the number of animals moved for each farm. The number of animals moved in any 14 day period is uniformly distributed over the VNTR 10 outbreak (as consistent with what is observed in the dataset of animal movements in Northern Ireland over the period investigated) creating a total of 7749685 movements. In each time step we move a constant number of cattle (19108) so that the total moved in the simulation is 7749685 using the following algorithm:

1. pick a farm-farm movement from the distribution of farm-farm movements, ignoring farms that are under movement restriction (see the paragraph on testing for details on these restrictions).

2. pick the number of cattle to be moved from the distribution of the number of animals moved off the departure farm.

3. perform pre-movement testing of all animals. If the departure farm contained infected animals we sample a number of these animals at random from a hypergeometric distribution and test them. If an animal fails the pre-movement test their SNPs are recorded as being sampled, they are removed from the simulation and the farm is put under movement restrictions with a follow-up test scheduled for 60 days. If none of the infected animals are detected they are moved to the new herd.

4. repeat until we have moved the required number of animals for the period.

Here we assume that the patterns do not change over timescales relevant to transmission and evolution or in a way that substantially influences the metrics used (i.e. the actual individual moves will be different, but not the overall pattern characteristics).

## Testing

Every head of cattle undergoes a pre-movement test with a sensitivity $\Omega$ and any that fail this is considered to be a 'reactor' (i.e. is in either the test-sensitive or infectious stage). Reactors are removed from the simulation and the herd they resided in is put under movement restriction; cattle are not allowed into or out of the herd until it passes two consecutive tests (scheduled at two monthly intervals starting from one month after the time of the breakdown). Each herd undergoes annual whole herd testing; herds that contain a reactor(s) are put under movement restriction and are required to pass two successive whole herd tests before being allowed to resume trading. Each herd is given a random test date at the start of the simulation which is repeated annually unless an infected animal is detected on the farm. Cattle that are found to be infected (in either the test sensitive or infectious states) are culled, i.e. removed from the simulation.

A record of all the transmission events that occur in our simulation is kept; thus we track all cattle→cattle, cattle→reservoir and reservoir→cattle transmission events. We allow for substitutions within the *M. bovis* genome at a rate of $\mu$ substitutions (single nucleotide polymorphisms, SNPs) per day. We don't track the actual loci that have evolved but rather label each strain of pathogen. Each transmission event is accompanied by a transfer of all the genetic information (i.e. the virtual pathogen containing all the SNPs) allowing us, over time, to calculate the distribution of single nucleotide polymorphisms (SNP) within the population. The same substitution rate is used for both cattle and within the reservoir.

## Seeding the model

We use the test histories of all herds observed to be part of the VNTR-10 outbreak and determine those animals that had a probability of being in an infected state at the start of the outbreak using the methods outlined in [1]. This identified 6 animals, all from the same farm, that harboured infection at the start of our simulation (June 1995) each of whom had the same probability of being in each infection state (S = 0.73, E = 0.0, T = 0.0, I = 0.27). We seed each simulation by setting the infection state of each of these animals according to these probabilities and assigning a unique set of SNPs to each infected cow. For each infected cow we seed the simulation with an infected reservoir animal with the same SNPs in one of the reservoirs connected to the farm (selected at random).

## Culling infected animals

Cattle are removed from our simulation according to the observed distribution of animal deaths; cattle are chosen at random, tested and if found to be positive the herd is placed under movement restriction and follow up tests are scheduled for 60 days (mimicking short interval testing used to "clear"a herd of infection. The herd remains under movement restriction until 2 successive clear tests are observed 60 days apart. We ignore animal births as we assume that calves are born free of the disease and enter the susceptible population that is not (explicitly) tracked.

## Hidden reservoir

We incorporated three different models for the hidden reservoir; none (there was no reservoir and the epidemic was driven by cattle to cattle transmission and movements only), considering both a single reservoir that is connected to every farm in the network, and reservoirs that had a radius of 2km (similar to the expected home ranges of badgers) so farms that were less than 4km apart could share one or more reservoirs (S2 Fig). Our dataset did not contain location data for those uninfected farms that were connected by movements so we assigned a reservoir to each of these but did not include overlaps with other reservoirs in the simulations. This resulted in a network with a subset of farms that were connected via a reservoir. For comparison, we also created a second network where the distribution of connected farms mimicked the distribution of connections that include farms for which we did not have locations. We refer to this as our "synthetic"network.

Several models of a hidden reservoir are compared in our simulations:

1. A single reservoir connecting every farm (giant reservoir model): infections into the reservoir are modelled as individuals without transmission within the reservoir. Since the reservoir connects all the farms this individual is free to re-infect any farm thus allowing for fast spatial transmission of *M. bovis* genotypes throughout Northern Ireland.

2. No shared reservoir: each farm has a reservoir that is not shared with any other. Infections into the reservoir are also modelled as individuals but without transmission within the reservoir. In this model, long range transmission of genotype information is only possible through movement of undiagnosed cattle.

3. Farms within 4km share a reservoir: to allow for a home range of wildlife we connect farms within a 4km range by a reservoir (S2 Fig). In this scenario we model the reservoir in three different ways:

    a. Individuals within the reservoir are not modelled (the maximum diversity model). In this case, whenever infection is passed from the reservoir into a herd, a new sequence

type is created with a number of accrued single nucleotide polymorphisms (SNPs or substitutions), counted from the point at which infection first entered the reservoir.

b. Individuals in the reservoir are modelled but without explicit transmission within the reservoir (the intermediate diversity model). When infection is transmitted from the reservoir into a herd an individual infection source is selected and their SNPs are transmitted along with any substitutions that might have occurred since they were infected.

c. Individuals in the reservoir are modelled but, without explicit transmission within the reservoir, and so diversity is only generated by the simulated cattle (the minimum diversity model). When infection is transmitted from the reservoir into a herd an individual is selected but no new SNPs are generated. This models the influence of a reservoir population which holds infection even when the local cattle population is clear but only maintains diversity via regular interaction with the cattle population.

4. The minimum diversity and maximum diversity models run on the synthetic network.

## SNPs

The number of samples taken from reactors shows a small but positive correlation with the time since the start of the outbreak (S3 Fig) so we sample from our transmission trees accordingly, sampling preferentially from the end of the outbreak to build a phylogeny from the underlying (simulated) epidemics. We use these trees to create the distribution of SNP differences between the tips of the tree (defined as $|(A \backslash B) \cup (B \backslash A)|$) where $A$, $B$ represent the sets of SNPs from the pair of samples.

The frequency distribution of the number of SNP differences is interpreted as a multinomial trial with $p_1, p_2, \ldots, p_n$, the probability of observing $1, 2, \ldots, n$, SNP differences and $x_1, x_2, \ldots, x_n$, the number of times we observed this number in our simulations. The distribution $p_1, p_2, \ldots, p_n$ is shown in Fig 2a. We can write our [partial] likelihood function as

$$\mathcal{L} = \frac{n!}{\Pi x_i!} p_i^{x_i} \tag{2}$$

where $n$ is to the total number of observed SNP differences.

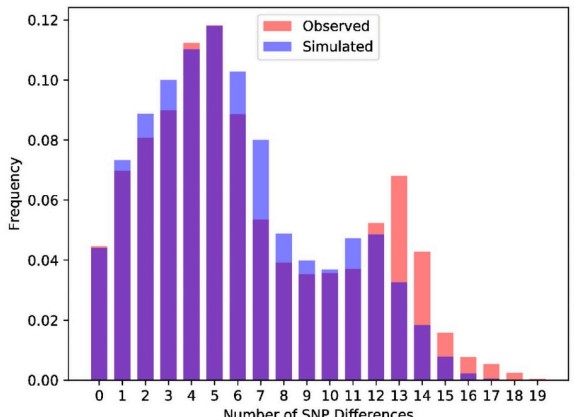
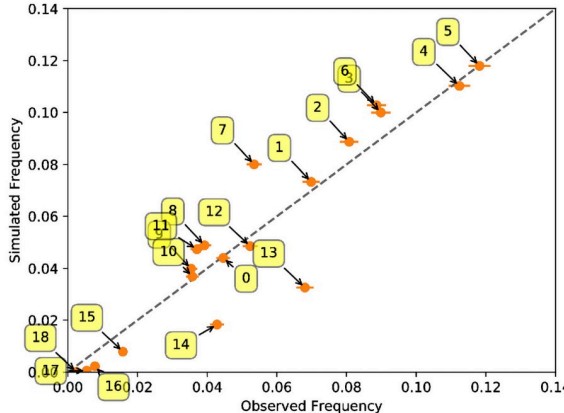

**Fig 2. Comparison of the pairwise distribution of SNP differences for observed cases in Northern Ireland and in our simulations.** The yellow boxes on the right hand plot indicate the number of SNP differences associated with each point.

## Simulating the model

2000 simulations are run from $15^{th}$ June 1995 until $31^{st}$ December 2010 (the period covered by our dataset) for each set of model parameters on a local Condor network. We record the transmission tree for each of the simulations. We calculate the expected value of the model as

$$E[\theta] = \langle \mathcal{L}(\mathcal{D}|\theta) \rangle \tag{3}$$

where the calculated likelihood is the mean of 2000 simulations of the model as described above. We perform a Markovian-random walk, where each parameter in the parameter set $\boldsymbol{\theta}$ is perturbed using a zero mean Gaussian random variable with a small variance, using the Metropolis-Hastings rejection algorithm to accept those parameter vectors, $\boldsymbol{\theta}$, that maximise the likelihood in Eq 2 to find the posterior values of $\boldsymbol{\theta}$ that corresponds to the maximum of the likelihood.

## Results

We compared our 5 different models that incorporated the pattern of recorded animal movements, births and removals. In each model investigated, we used the same priors (Table 1) which were chosen on the basis of existing field and experimental estimates [14–17] where they existed. The substitution or SNP creation rate is informed by our previous estimate [3]. For the transmission rates, rate of the pathogen and test sensitivity, we used non-informative priors, i.e. uniform priors with a large range.

We calculated the AIC score of each of our 5 models to determine which model best describes the observed diversity (for our final set of models presented here, each model contains the same number of parameters so in comparing the AIC scores of each model we are indirectly comparing their log-likelihoods). According to the AIC scores the preferred model that best described the observed data was one that generated no additional diversity within the reservoir population (the minimal diversity model)(Table 2).

Considering the univariate parameter distributions in the posterior (Table 3), the length of the exposed stage (i.e. $1/\sigma$) was estimated to be 14.3 days (with lower and upper quartiles 5.5–16.1 days) in line with previous estimates (60 hours-100 days) [1, 7]. This is towards the longer duration of the prior for this value. The length of the test-sensitive stage (i.e. $1/\gamma$) was estimated to be 155.7 days (with lower and upper quartiles 147.1 and 169.5 days, respectively) also similar to previously published estimates of 180 ± 20 [1, 7, 14, 18]. Estimates for the sensitivities of the

**Table 1. Summary of the priors used in the model.** In the cases of $\beta$, $\alpha_{RC}$, $\alpha_{CR}$, $\mu$ we used non-informative priors whereas in the case of $\sigma$, $\gamma$, $\Omega$ the priors were chosen on the basis of existing field and experimental estimates [3, 15, 16].

| Description | Sampling Distribution |
|---|---|
| Cattle-cattle transmission rate, $\beta$ | Uniform$[1 \times 10^{-6}, 1 \times 10^{-1}]$ (individuals $\times$ time)$^{-1}$ |
| Rate exposed cattle become test sensitive, $\sigma$ | Uniform[6hours–100days] time$^{-1}$ |
| Rate test sensitive cattle become infectious, $\gamma$ | Uniform[4months–11months] time$^{-1}$ |
| Probability that a detectable animal is detected, $\Omega$ | Uniform$[0.4 - 0.8]$ |
| Reservoir-cattle transmission rate, $\beta_{RC}$ | Uniform$[1 \times 10^{-3}, 0.4]$ (individuals $\times$ time)$^{-1}$ |
| Cattle-reservoir transmission rate, $\beta_{CR}$ | Uniform$[1 \times 10^{-3}, 0.4]$ (individuals $\times$ time)$^{-1}$ |
| SNP generation rate per day, $\mu$ | Uniform$[0.00001, 0.005]$ time$^{-1}$ |

**Table 2. AIC scores for the various models investigated.** The best model is one in which we have a reservoir population that does not actively contribute to pathogen diversity the minimal diversity model).

|  | AIC Score |
|---|---|
| No Connecting Reservoirs | 4370.8 |
| Intermediate Diversity model 2km radius Reservoir | 3816.2 |
| Maximum Diversity model 2km radius Reservoir | 3232.2 |
| Giant Connecting Reservoir | 3056.8 |
| Minimal diversity model 2km radius Reservoir | 2192.7 |

**Table 3. Summary of the posterior estimates for the minimal diversity model which best fits the data from our calculations, (see Table 2 for the AIC scores for each model).** 95% credible intervals are given in square braces.

| Description | Posterior Estimates |
|---|---|
| Cattle-cattle transmission rate, $\beta$ | $6.3 \times 10^{-6}$ $(\text{individuals} \times \text{time})^{-1}$ $[4.4 \times 10^{-6}, 7.0 \times 10^{-6}]$ |
| Rate exposed cattle become test sensitive, $\sigma$ | $0.042 \text{ time}^{-1} \approx 23.8$ days $[0.041, 0.062] \approx [16.1, 24.4]$ days |
| Rate test sensitive cattle become infectious, $\gamma$ | $0.0062 \text{ time}^{-1} \approx 161.3$ days $[0.0052, 0.0064] \approx [156.3, 192.3]$ days |
| Probability that a detectable animal is detected, $\Omega$ | 0.506 [0.410, 0.626] |
| Reservoir-cattle transmission rate, $\beta_{RC}$ | $4.39 \times 10^{-6}$ $(\text{individuals} \times \text{time})^{-1}$ $[4.01 \times 10^{-6}, 6.73 \times 10^{-6}]$ |
| Cattle-reservoir transmission rate, $\beta_{CR}$ | $2.45 \times 10^{-6}$ $(\text{individuals} \times \text{time})^{-1}$ $[1.28 \times 10^{-6}, 5.06 \times 10^{-6}]$ |
| SNP generation rate per day, $\mu$ | $0.0041 \text{ time}^{-1}$ $[0.0033, 0.0044]$ |

whole herd test test, 59% (with a 95% credible interval of 52–63%), are also consistent with previous observations (50%-100%) [15–17] and simulations (36%-55%) [7, 18]. Neither the minimum or maximum diversity model on our synthetic network generated a stable posterior estimate (not shown).

The best fit model is compared to the SNP distance histogram in illustrated in Fig 2a and 2b, showing considerable fidelity of the model to these data; the simulated distribution displays both the bi-modal nature of the differences and matches the observed distribution. Though there is evidence of a small amount of overdispersion for large SNP differences while underestimating lower numbers of SNP differences, results lie within the 95% CI's for most SNP differences compared. By comparison, a visual inspection for the other models shows that, despite very similar univariate parameter estimates (S4 Fig), SNP distributions were clearly inferior when comparing to the data (Fig 3a–3d)).

In order to determine whether removal of the reservoir alone could result in eradication of bTB in this system, we sample from the posterior distributions of our parameters and use this to calculate the cattle-only $R_0$ using a next generation matrix approach [19, 20]. $R_0$ was found to have a mean value of 1.89 with a standard deviation of 1.04, as in S5 Fig. Thus the most likely result across the posterior is that, in the absence of testing, cattle-to-cattle transmission can maintain bTB on its own (i.e. $R_0^{cattle} > 1$).

## Discussion

The development of model inference with dense data from one mammalian host, but only sparse, or in this case no, data from another is a challenging one. We discuss here a

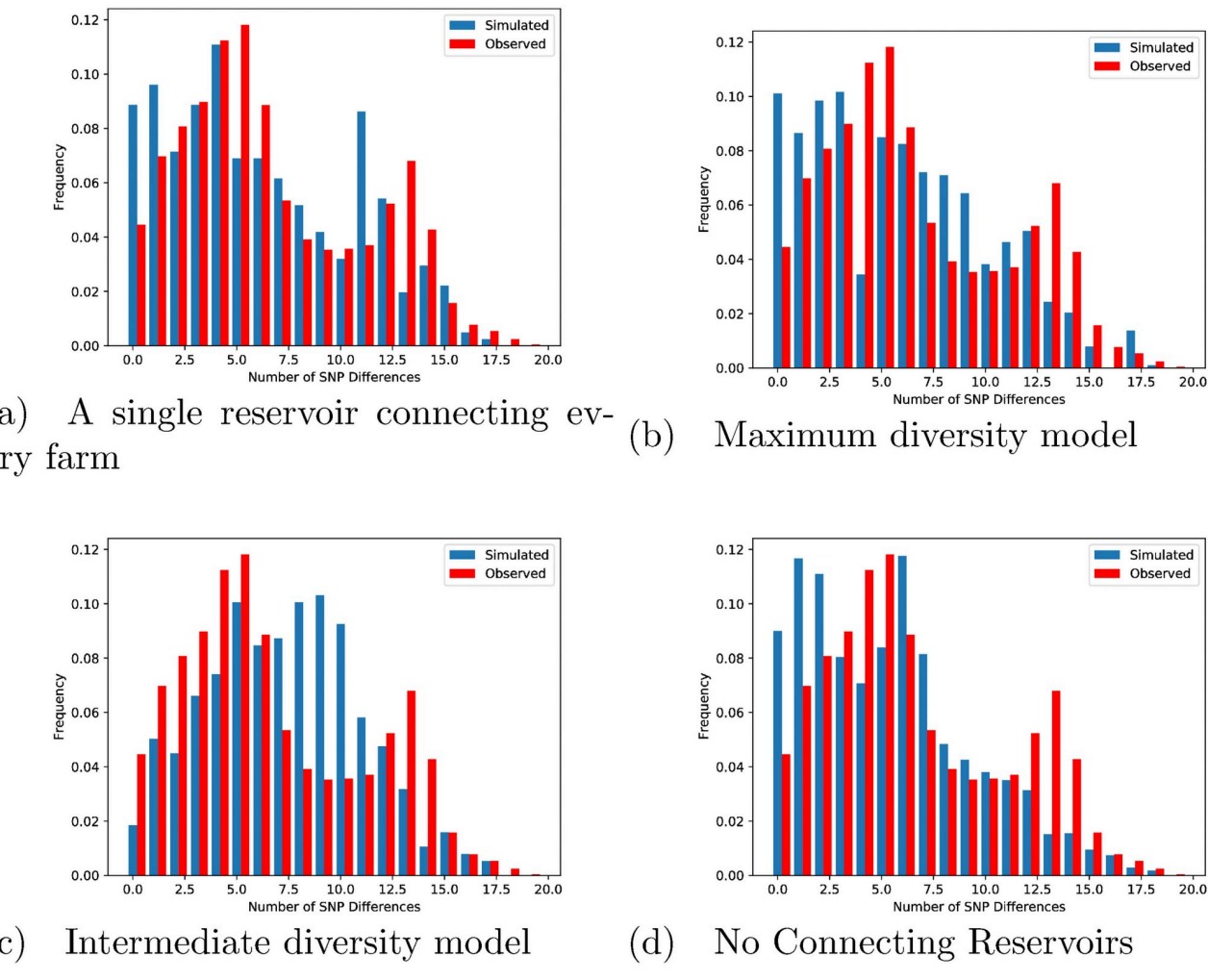

**Fig 3. Comparison of the pairwise distribution of SNP differences for the four rejected models.** (a) A single reservoir connecting every farm (top left), (b) Maximum diversity model (top right), (c) Intermediate diversity model (bottom left), (d) No Connecting Reservoirs (bottom right).

generalisable approach to doing so, exploiting the existence of spatio-temporal signatures in the phylogeny that are inconsistent with the pattern of known recorded movements and contacts amongst the densely observed host (in this case, the cattle). An important question is whether or not our reservoirs are in fact infected badger populations. In our view this is the most parsimonious explanation. First, the poor fit of the cattle-only model suggests that the transmission mechanisms involved are driven by wholly local, unrecorded interactions that have a stronger impact than the direct movement of cattle between farms. Thus any hidden agent infecting the cattle would have to be spatially constrained in a way that the infected cattle that are allowed to move between farms are not—i.e. behave very much like badgers. Second, our analysis supports the need for a reservoir that is a source of infection but does not generate substantial new genetic diversity, providing further hints of the nature of this reservoir. This is consistent with recent results from an analysis of cattle and badger derived *M. bovis* samples also from Northern Ireland where preliminary results indicate that, while badgers do contribute to cattle transmission for that study site they did so rarely and also rarely infect each other [21]. Biologically, this could be interpreted as implying regular exchanges between the cattle and badger populations with infected badgers "returning" very similar genetic types (in the

model, the same type) to those it has recently been infected with by the cattle—like our minimum diversity model. In contrast, the maximum or intermediate diversity models would imply a badger population that is able to maintain diversity independently. While this has been observed elsewhere [6], the results here highlight the differences that arise from local geographic and management considerations [22], even though the scenarios are superficially very similar. Despite this consistency we must allow for other plausible mechanisms that may also contribute, including e.g. unrecorded or illegal movements of cattle, shared pastures, unrecorded movement of bulls etc.—this could be investigated via field investigations.

*A priori*, it might be expected that our synthetic network, which provides estimated geographical locations for all available herds, would be a better representation of the true epidemiological situation. Thus our inability to fit this model to the data is somewhat surprising. However, previous analyses of annual testing areas in England identified having a past history of outbreaks of bTB as being the most important risk factor predicting future outbreaks [23, 24], implying the existence of important farm-level risk factors not captured in our data, and thus more consistent with a model where only herds that have been infected at some points in our records have this localised risk.

Our metric is based solely on observations of genetic diversity and do not incorporate epidemiological observations. While our results show that this alone is sufficient to distinguish between importantly different models of transmission, our estimate of transmission parameters depends on the ability to generate diversity. Because in our preferred (minimum diversity) model the reservoir generates no independent diversity, the ability of the model to infer transmission rates from the reservoir is limited (S5 Fig). Ongoing work involves extending the partial likelihood approach to include these epidemiological observations, better exploiting the detailed contact information in the movement data.

Despite its limitations, the ability of our approach to identify key factors with heavily biased data is promising. As this will often be the case where different species are involved in transmission and pathogen replication, our method points a way to generate important insights about hidden reservoir populations.

## Supporting information

**S1 Fig. Schematic representation of the model used.** The code implementing this model can be found at https://github.com/anthonyohare/NIBtbClusterModel.
(EPS)

**S2 Fig. Relative locations of the herds from which VNTR-10 samples were taken (km scale).** All herd locations are translated to anonymise them. Herds within a 2km radius are assumed to be connected by a wildlife reservoir (left) while the other herds are joined, indirectly, through animal movements (uniformly distributed throughout the length of the simulation and between farms) or genetically (right). The determination of those herds that had a seperation of 4km was made before the location data was transformed.
(EPS)

**S3 Fig. Number of reactors discovered with the number of samples that were successfully grown and analysed.**
(EPS)

**S4 Fig. Posterior kernel density estimates for the parameter distributions for the model.** Here $\beta$, $\sigma$, $\gamma$ are the transition rates in our model, $\Omega$ is the sensitivity of the routine and abattoir tests, $\beta_{CR}$, $\beta_{RC}$ are the transmission rates from cattle to reservoir and reservoir to cattle respectively and $\mu$ is the number of new SNPs generated in the model per day. The horizontal scale

on each figure corresponds to the priors used and were taken from existing field and experimental estimates [15, 16]. In the case of the $\sigma$ parameter our modelling suggests that the length of time an animal is in this stage is towards the longer prior estimate.
(EPS)

**S5 Fig. Variation of cattle-only $R_0$ with external force of infection where each dot represents a simulation.** The dashed line is the $R_0 = 1$ line, indicating the proportion of the posterior where removal of the external force of infection entirely would result in eradication of the disease (i.e. the proportion below $R_0 = 1$), our simulations suggests this proportion is $\sim 0.068$. The dotted line is the mean value of the distribution of $R_0$ in our simulations and decreases slowly with increasing external force of infection (the slope of this line is small but negative, -0.083) indicating that $R_0$ decreases slowly with increasing external force of infection.
(EPS)

**S1 Table. Comparison of the mean posterior values for the parameters in each of the 5 models.** Minimum diversity model (Min), Intermediate diversity model (Int), Maximum diversity model (Max), A single reservoir connecting every farm (Giant), No Connecting Reservoirs (None).
(PDF)

## Acknowledgments

We thank Dr. Georgina Milne at AFBI-NI, who provided some corroborating analyses for the distribution of herd geo-locations using data unavailable to us in this project.

## Author Contributions

**Conceptualization:** Anthony O'Hare, Stanley McDowell, Robin A. Skuce, Rowland R. Kao.

**Data curation:** David M. Wright, Carl McCormick.

**Formal analysis:** Anthony O'Hare, David M. Wright.

**Funding acquisition:** Rowland R. Kao.

**Investigation:** Carl McCormick, Hannah Trewby, Rowland R. Kao.

**Methodology:** Anthony O'Hare, Daniel Balaz, Rowland R. Kao.

**Project administration:** Robin A. Skuce.

**Software:** Anthony O'Hare, Daniel Balaz.

**Supervision:** Robin A. Skuce, Rowland R. Kao.

**Validation:** Daniel Balaz.

**Visualization:** Anthony O'Hare.

**Writing – original draft:** Anthony O'Hare, Rowland R. Kao.

**Writing – review & editing:** Anthony O'Hare, Rowland R. Kao.

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
