## [Decision Letter · Decision Letter 0]

28 Dec 2020

Dear Professor Kao,

Thank you very much for submitting your manuscript "A new phylodynamic model of Mycobacterium bovis

transmission in a multi-host system uncovers the role of the

unobserved reservoir." for consideration at PLOS Computational Biology.

As with all papers reviewed by the journal, your manuscript was reviewed by members of the editorial board and by several independent reviewers. In light of the reviews (below this email), we would like to invite the resubmission of a significantly-revised version that takes into account the reviewers' comments.

We cannot make any decision about publication until we have seen the revised manuscript and your response to the reviewers' comments. Your revised manuscript is also likely to be sent to reviewers for further evaluation.

Sincerely,

Miles P. Davenport, MB BS, D.Phil

Associate Editor

PLOS Computational Biology

Nina Fefferman

Deputy Editor

PLOS Computational Biology

Reviewer's Responses to Questions

**Comments to the Authors:**

Reviewer #1: I really enjoyed reading this paper. The authors take a novel computational approach to assessing the potential role of an unsampled reservoir population in creating the observed SNP diversity of Mycobacterium bovis in cattle. I support this paper for eventual publication but found that there were a number of components that needed more details.

1. The authors provide details on the model structure as text. I don't think this is sufficient to be able to repeat any of there analyses or build off of their work here and easily apply this work to another system. Providing all, or some key parts, of the code would address this. I also wonder if you could include a mock figure indicating where SNPs are generated in the different scenarios.

2. The likelihood only addresses the SNP data, but not any of the epidemiological data. Yet the authors clearly have the movement information and whether positive cattle herds are linked by a prior movement event or not, which would be indicative of a wildlife reservoir as well.The authors mention this around line 273, but I think more could be said here. Given what the authors have accomplished here, what are the pros/cons and difficulties that would be involved in trying to also incorporate more of the epidemiological data.

3. Figure 6 illustrates how some of the posteriors are bumping up against the edges of the uniform prior distributions. This warrants some discussion, if not a complete re-run with wider priors on some of these. This may be an issue of computational time, but that also wasn't well described.

4. I needed more help relating some of the modeling scenarios to biological systems. For example, how would you have a wildlife reservoir that does not generate any SNP diversity (the best model of low diversity). This would seem to imply no transmission within the wildlife species and all transmission is contained in the spillover process. If true, then I'm not sure reservoir is the right term for the wildlife population. Could these scenarios be related to Haydon et al 2002's circles and squares diagram? And does this scenario imply that R0 is less than one in the badger population alone, otherwise one would acrue SNP diversity in the wildlife population? How does this conclusion match with the other badger-TB work?

Minor points.

Spell out Northern Ireland for those less familiar with that region of the world.

I was confused by the legend on Fig. 3. It states that R0 <1 indicates the proportion of the posterior where the removal of the external force would result in disease eradication. I would have thought that if R0 <1 then that indicates eradication without any need for additional removal of the external force of infection. Maybe this gets to an earlier point about whether badgers maintain the disease in the absence of cattle.

Reviewer #2: In the paper "A new phylodynamic model of Mycobacterium bovis transmission in a multi-host system uncovers the role of the unobserved reservoir" , O'hare et al use mathemathical compartment modeling to explore how hidden reservoirs contribute to the observed bTb circulation in Northern Ireland. It is an interesting and well-written paper that deals with a hot topic - The role of badgers in bTb spread. (And could therefore be used in this ongoing debate about the notorious control work that has been done in the UK). The paper concludes that the best-fitting model (among the 5 models tested) is one that ascribes a central role to both cattle-to-cattle transmission but with the addition of a hidden reservoir that can infect local farms. Interestingly, the paper offers no data to support the idea that this hidden reservoir consists of badgers, although it is heavily implied.

Below follows some specific comments to the paper:

- The link to badgers is not clear. Although badgers are not mentioned in the title, they are repeatedly mentioned in both the abstract and the main text. It is mentioned in the "Data" chapter that whole-genome sequencing of roadkilled badgers have been undertaken, although these data were not used to inform the simulations. Why not? Badger sequences also appear in Figure 1, and from this figure it is clear that at least some badger-originating sequences are closely related to cattle. I worry that this paper will be seen as demonstrating the role of badgers specifically, and while the hidden reservoir may indeed consist of badgers, there are many other plausible explanations, some of which the authors themselves discuss, like untracked movements, shared pastures, bulls etc.

- Figure 1 - It is unclear to me why this figure is in the paper. It is taken directly from Trewby et al, 2015, and does not seem particularly relevant for the simulations except to demonstrate the basis for the SNP distribution.

- I'm not 100% sure I understand the SETI model on page 4. Is there a separate process that seeds infection to the reservoir and the reservoirs are modeled with separate ODEs? I assume there's also a process for culling of infected that's not specified in this model? On page 6 movement restrictions upon a positive test is described, but it says nothing about test-positive animals being culled.

- On page 6-7, the 5 different simulation setups are described. It would be helpful if the authors could use this terminology when they refer to specific scenarios elsewhere in the paper (e.g. 3A). In fact I am confused as to which model ended up as the winner in the end. On page 8 the most parsimonious is the model where "that generated NO additional diversity within the reservoir population 3". In the table on the following page (with AIC scores), a low-diversity model is prefered. Later on the same page the best model is one that "does not generate substantial new genetic diversity", while again on page 10 "our preferred model incorporates a reservoir which generates NO independent diversity".

- The mutation rate prior and posterior seem extremely high for bTb. In my experience they are much closer to the 1.0E-7 mark if we are talking about the whole-genome rate. Is this rate for some smaller set, eg just the complement of SNP sites?

- On the very last sentence of the results is an inverted question mark. - Should be a "greater than" sign?

- I can understand that putting potentially identifiable movement data online can be impossible under EU law. Would the authors at least consider sharing the code they used in their simulations?

- Figure 6 - The scale of sigma is too large to see anything. What is the unit of sigma? This might be obvious from table 2 but I can't figure it out. (0.042 = 23.8 days ?)

Reviewer #3: In this work, authors present a coupled dynamical model where the dynamics of bovine tuberculosis (bTB) transmission through cattle in Northern Ireland, between 1995 and 2010 is described. The model integrates three dynamical processes (namely, bTB dynamics, cattle movement between farms and genetic diversity dynamics in the pathogen); and its main objective is to provide a modeling framework through which the genetic diversity patterns that are observed among the registered outbreaks across the territory during the mentioned period can be explained only as a consequence of incorporating to the models the interaction between herds and wildlife reservoirs that are carefully modelled, in this case Eurasian badgers.

I think the work is appealing, and the modeling framework presented is comprehensive, and a valuable tool to understand the contribution of wild reservoirs to the control of this important economical and public health problem.

I have, however a series of specific concerns that are mostly related to the way that the model is described (which I think, in some aspects is a bit cumbersome, and/or leaves important questions unaddressed), as well as the way some of the results are presented. If authors managed to provide an answer to these questions, I would be happy to recommend their work for publication in the journal.

1. Methods: the explanation of the algorithm provided in the text within the methods section is quite cumbersome at times. I would suggest authors to try to structure more explicitly the information about the three dynamical processes coupled in their model (i.e. infection dynamics, strains evolution and cattle movements). Similarly, I strongly suggest authors to include a global algorithm summary as well.

2. In particular, I found confusing the description of how do the model does two things that are tightly related to each other, that is, keeping track of the total amount of susceptible individuals, as well as the total amount of individuals in each herd. Specifically, it would seem that authors introduce, at each time step, corrections in the population of each herd, to make the evolution of herd sizes compatible to the data. However, I see that this can be done in at least three ways, and it is not clear to me which one authors used:

a. The total amount of susceptible animals during the simulations is in fact the variable whose time evolution is to be “fitted” to the data of herd size, neglecting the contribution to herd size of infected animals. Therefore, the animals added/removed to simulate births/deaths are all susceptible.

b. The total amount of animals, both susceptible and infected, during the simulations is in fact the variable whose time evolution is to be “fitted” to the data of herd size. Here, the animals added/removed to simulate births/deaths could be:

b.1 all susceptible.

b.2 Distributed between infection classes proportionally to their prevalences in each herd, at each time.

My understanding is that authors chose option a, but I cannot be sure, reading the text. In any case, they should state explicitly how this was done, and, if their choice implies neglecting the contribution of infected individuals to the total herd size, it would be helpful to see, for example, the distribution of the fraction of infected individuals across herds/and or time, which would be very helpful for the reader to get a good grasp of how sensible that simplification may be.

3. I miss, also in the methods section, a diagram reuniting the info of how many animals, herds, movements between herds, herd breakdowns, isolates, SNPs, simulations, reservoirs, etc. are being simulated.

4. How is the movement distribution defined? The usage of the word distribution seems to imply that authors built a frequency distribution of movements as a function, for example, of the distance between farms, and draw movements from this distribution stochastically, in their simulations. This would allow movements among all farms, including those that were not involved in cattle exchange according to the data. Is this correct? It could also be that movement frequencies (propensities) are proportional to some other variable, such as herd sizes of receptor and/or sender herds, or proportional to the frequency of trade of each farm as observed in the dataset. In any case, how are movement propensities written down within the Gillespie algorithm? Do they allow connecting farms for which no cattle movement was recorded in the dataset, via cattle movements during the simulations?

5. I have two doubts concerning the assignment of reservoirs to herds according to model 3:

5.1 In lines 43-44, we read that data anonymization implies scaling of distances (plus rotation and translation). How is it possible, then, to know which farms are within a 4 km range in order to decide whether it makes sense or not to assign them a single badger reservoir?

5.2 In lines 102-109 we read that no shared reservoir was included for the farms without movements. If that is the case (especially if the answer to the last question in point 4 is positive), this would introduce a bias in the simulations, which would generate more disease prevalence in the farms for which we had location data, bc they, by construction, can share reservoirs. An easy, alternative null model for these farms where location data is unavailable would be that the proportion of farm groups of size=2,3, etc. sharing a common reservoir should match that of the data for which we do have farms´ locations. If authors think this should no introduce substantial modifications, I think this should be discussed, and properly justified.

6. Why is it reasonable to neglect transmission within the reservoirs? This question should be discussed in the text.

7. How is the metropolis-Hasting algorithm set up? What is the objective function? (e.g. is it based on matching the pairwise distributions of SNP differences, or does it include the observe incidence?)

8. Does table 2 refer to the best-likelihood model (model 3)? Should be stated in the table header.

9. Do the model compared through the AIC differ in the number of parameters? If so, how? If the vector theta of parameters only include the parameters listed in the tables 1 and 2, then all models have the same number of parameters, and comparing AIC´s means comparing likelihoods.

10. How is the external force of infection obtained? Is there an statistical test that can support the claim that “R0 decreases slowly with increasing external force of infection”?, read in lines 253-254? (If not, that claim should be removed). I believe that the discussion around figure 3 is remarkably insufficient, and that this figure requires more explanation (I have to admit that I could not get to understand what is the point the autors try to make here). From a data vis. Point of view, very elementary questions are pertienent: what do points represent? Individual simulations, perhaps? How many dots are under R0=1?, What does the blue dashed line represent (the mean r0, perhaps?). I feel that, if the point here is to interrogate for the ability of the model to produce observed incidence levels, it would be desirable to also have a figure where observed vs simulated incidence is compared (see comment 12 below).

11. In the parenthesis at the end of line 256 there is a typo.

12. In figure 2 and table 2, authors present the results of their best-fit model. I think that, to be sure that the model 3 is really the best one (and that it reproduces coherently the data observed), there is information that is still missing. In my opinion, it would be important to include information about how different the predictions (e.g. likelihoods, observed vs simulated incidences, and pairwise distribution of SNPs differences) compare across the five models studied.

**Have all data underlying the figures and results presented in the manuscript been provided?**

Reviewer #1: **No: **It appears that some of the data cannot be made available to the public.

Reviewer #2: **No: **Epidemiological / demographic farm data and livestock movement data have not been provided. (As stated by the authors due to EU confidentiallity laws)

Reviewer #3: Yes

PLOS authors have the option to publish the peer review history of their article (what does this mean?). If published, this will include your full peer review and any attached files.

Reviewer #1: No

Reviewer #2: No

Reviewer #3: No
---

## [Decision Letter · Decision Letter 1]

18 Apr 2021

Dear Professor Kao,

Thank you very much for submitting your manuscript "A new phylodynamic model of Mycobacterium bovis

transmission in a multi-host system uncovers the role of the

unobserved reservoir." for consideration at PLOS Computational Biology. The reviewers were happy that their concerns had been addressed, although one had some additional suggestions for your consideration.

Please prepare and submit your revised manuscript within 7 days. If you anticipate any delay, please let us know the expected resubmission date by replying to this email.

When you are ready to resubmit, please upload the revised manuscript.

Sincerely,

Miles P. Davenport, MB BS, D.Phil

Associate Editor

PLOS Computational Biology

Nina Fefferman

Deputy Editor

PLOS Computational Biology

[LINK]

Reviewer's Responses to Questions

**Comments to the Authors:**

Reviewer #1: The authors have addressed my concerns from the previous review.

Reviewer #3: The resubmitted version of the article essentially addresses all the questions I raised in my first revision. I think that the present version of the manuscript reads (much) better, it is better self-contained, and more accessible for the general reader.

Therefore, it is my pleasure to recommend the work for publication, and I only have three certainly minor points to raise, which I of course do not think should require further revision:

1. In line 40, we read "58 herds", perhaps that number should read 52?

2. In lines 256-263, authors cite previous works claiming that the estimates of the parameters found through their modelling approach are similar to previous estimates. I think that it would be extremely useful for the reader to get an idea of the quantitative potential of the approach to include in the text the estimates of the parameters that are produced or reported in these references.

3. In table 1; the units of gamma parameters ("months per day") and sigma parameters ("hours, or days per day"), are confusing. In general, according to the ODEs in eqs. (1), beta dimensions are (time * individuals)^-1, while sigma and gamma dimensions are time^-1.

**Have the authors made all data and (if applicable) computational code underlying the findings in their manuscript fully available?**

Reviewer #1: Yes

Reviewer #3: None

PLOS authors have the option to publish the peer review history of their article (what does this mean?). If published, this will include your full peer review and any attached files.

Reviewer #1: No

Reviewer #3: No

Figure Files:

Data Requirements:

Reproducibility:

References:

---

## [Editor Report · Decision Letter 2]

25 Apr 2021

Dear Professor Kao,

We are pleased to inform you that your manuscript 'A new phylodynamic model of Mycobacterium bovis

transmission in a multi-host system uncovers the role of the

unobserved reservoir.' has been provisionally accepted for publication in PLOS Computational Biology.

Best regards,

Miles P. Davenport, MB BS, D.Phil

Associate Editor

PLOS Computational Biology

Nina Fefferman

Deputy Editor

PLOS Computational Biology

---

## [Editor Report · Acceptance letter]

16 Jun 2021

PCOMPBIOL-D-20-01334R2 

A new phylodynamic model of Mycobacterium bovis
transmission in a multi-host system uncovers the role of the
unobserved reservoir.

Dear Dr Kao,

I am pleased to inform you that your manuscript has been formally accepted for publication in PLOS Computational Biology. Your manuscript is now with our production department and you will be notified of the publication date in due course.

With kind regards,

Zita Barta
